# Predictive Value of Maternal HbA1c Levels for Fetal Hypertrophic Cardiomyopathy in Pregestational Diabetic Pregnancies

**DOI:** 10.3390/children12030312

**Published:** 2025-02-28

**Authors:** Angel Chimenea, Ana María Calderón, Guillermo Antiñolo, Eduardo Moreno-Reina, Lutgardo García-Díaz

**Affiliations:** 1Department of Materno-Fetal Medicine, Genetics and Reproduction, Institute of Biomedicine of Seville (IBIS), Hospital Universitario Virgen del Rocio/Consejo Superior de Investigaciones Científicas (CSIC)/University of Seville, 41004 Seville, Spain; angel.chimenea.sspa@juntadeandalucia.es (A.C.);; 2Centre for Biomedical Network Research on Rare Diseases (CIBERER), 41013 Seville, Spain; 3Fetal, IVF and Reproduction Simulation Training Centre (FIRST), 41010 Seville, Spain; 4Department of Gynecology, Hospital Universitario Virgen del Rocio, 41013 Seville, Spain; 5Department of Surgery, University of Seville, 41004 Seville, Spain

**Keywords:** diabetes mellitus, hypertrophic cardiomyopathy, obstetric outcomes, maternal health

## Abstract

(1) Background: This study investigated the utility of first-visit HbA1c levels as a predictor of fetal hypertrophic cardiomyopathy (FHCM) in women with pregestational diabetes mellitus (PGDM). (2) Methods: A retrospective observational cohort study was conducted among all pregnant women with PGDM between 2012 and 2019. (3) Results: Of 329 participants, 5.8% had fetuses diagnosed with FHCM. These women exhibited significantly higher pregestational HbA1c (8.2% vs. 7.3%, *p* = 0.003) and higher first-visit HbA1c (7.6% vs. 6.9%, *p* = 0.001) and were less likely to have planned their pregnancies (*p* = 0.035). Fetuses with FHCM demonstrated a higher incidence of macrosomia (63.2% vs. 17.7%, *p* < 0.001; OR 9.20, 95% CI 3.31–25.58). Receiver-operating characteristic (ROC) analysis indicated an adequate predictive capacity for FHCM using first-visit HbA1c (AUC 0.75). An HbA1c threshold of 7.15% provided the best discriminative power, encompassing 38.9% of the cohort. (4) Conclusions: These findings underscore the value of assessing first-visit HbA1c levels for predicting FHCM in women with PGDM. The significant association between glycemic status and FHCM highlights the importance of optimizing glycemic control before and during pregnancy. Establishing optimal HbA1c cutoffs enables effective risk stratification and supports targeted clinical interventions.

## 1. Introduction

Pregestational diabetes mellitus (PGDM) is a chronic disease that has long-term implications on maternal micro and macrovascular health, significantly impacting pregnancy outcomes [1,2,3,4,5,6,7]. Currently, 60 million women of reproductive age worldwide have diabetes mellitus (DM), with projections indicating a doubling of this number by 2030 [8].

PGDM is associated with a significantly higher risk of various types of congenital malformations, while some types show more than a tenfold increase compared to non-diabetic pregnancies [9]. Moreover, there is evidence that pregestational diabetes markedly increases the risk of stillbirth, and this risk grows with higher HbA1c levels. Poor glycemic control is also related to large-for-gestational-age (LGA) fetuses and stillbirth [10]. The most common congenital malformations are cardiac anomalies, affecting the fetal heart structurally and functionally [11,12].

Fetal hypertrophic cardiomyopathy (FHCM) is a condition characterized by abnormal thickening of the heart muscle in the absence of abnormal conditions (such as hypertension or fetal valvular disorder) or systemic illness, with alterations in myocardial composition [13,14]. The incidence and prevalence of FHCM in both the general obstetrics population and in diabetic pregnant women are poorly defined due to many cases being asymptomatic or oligosymptomatic at birth [15]. The incidence of FHCM ranges from 44% in diabetic pregnancies (including asymptomatic cases) [14,16] to 13% when considering only clinically expressed cases [14].

The etiology of FHCM is identifiable in approximately 50% of cases, primarily associated with fetal hyperinsulinism resulting from maternal PGDM and the related oxidative stress [15,17]. Given its relationship with hyperinsulinemia and metabolic control, it is critical to define a patient profile with a higher risk of FHCM occurrence from the early stages of pregnancy to optimize monitoring and enhance the possibility of prenatal diagnosis.

The aim of this study is to establish the relationship between metabolic control at the beginning of pregnancy, defined by the HbA1c level at the initial prenatal visit in the first trimester of pregnancy, and the occurrence of FHCM. As a secondary objective, we aim to determine the predictive power of this parameter as well as optimal cutoff points.

## 2. Materials and Methods

### 2.1. Study Design and Outcomes

In this retrospective observational cohort study, we investigated the predictive value of HbA1c levels at the initial prenatal visit in the first trimester of pregnancy for the development of FHCM in pregnant women with PGDM. We categorized the study participants into two main groups: mothers with fetuses diagnosed with FHCM and another comprising fetuses without FHCM. Parameters potentially linked to the occurrence of FHCM were determined across various domains: biometric parameters (weight at the beginning and end of gestation, BMI at the beginning and end of gestation), analytical parameters (HbA1c at pregestational visit, first prenatal visit, last prenatal visit, postpartum, and increment during gestation), and diabetological factors (pregnancy planning, White classification, insulin dosage at the beginning and end of gestation, total insulin increment). Subsequently, we assessed the overall performance of HbA1c at the first visit for predicting FHCM.

We identified the study cohort through a prospectively compiled, anonymized database of pregnant women diagnosed with PGDM who were managed at the Diabetes and Pregnancy Clinic (Maternal–Fetal Medicine, Genetics, and Reproduction Unit) of Virgen del Rocío University Hospital (Seville, Spain), between 2012 to 2019.

All follow-up appointments were conducted at the Diabetes and Pregnancy Clinic, following a standardized protocol specifically designed for patients with PGDM. This protocol was developed based on guidelines established by the Spanish Society of Obstetrics and Gynecology (SEGO), and is summarized in Table 1.

### 2.2. Diagnosis of FHCM and Echocardiographic Parameters

The follow-up echocardiogram was performed between 28 and 32 weeks of gestation, when hyperinsulinemia and consequent oxidative stress are most likely to manifest and reveal pathological changes in fetal cardiac structure and function [18]. The diagnosis of FHCM was primarily based on interventricular septal thickening exceeding two standard deviations (SD) relative to the mean for gestational age [18] (Figure 1) and a modified myocardial performance index (MPI) higher than 0.43 [18]. Normal reference values for septal thickness were defined according to the standards determined by Schneider et al. [19] using the Fetal Medicine Barcelona online calculator [20]. Additional findings, including tricuspid regurgitation, a shortened and narrower left ventricle, and altered myocardial deformation patterns, were also considered supportive evidence of FHCM [21,22]. Some authors recommend the use of tissue and spectral Doppler to further assess fetal cardiac function [23], although these techniques were not employed as definitive diagnostic criteria in this study. According to recent data, an appropriately timed echocardiogram between 28 and 32 weeks of gestation can detect up to 96.4% of FHCM cases [12].

### 2.3. Participants and Eligibility Criteria

The eligibility criteria encompassed pregnant women with confirmed diagnoses of FHCM who underwent prenatal care at our institution, as well as those with singleton pregnancies. Exclusion criteria included patients with incomplete pregnancy-related data collection, multiple gestations, or gestational DM. Additionally, patients whose medical records were unavailable or insufficiently documented, as well as those who had not completed their pregnancy (e.g., induced abortion), were excluded from this study.

### 2.4. Data Compilation and Analysis

Data were collected from patient and neonatal medical records, and statistical analyses were conducted using the IBM Statistical Package for the Social Sciences (SPSS) version 25 (IBM Corp., Chicago, IL, USA). Quantitative variables were expressed as mean (central tendency) and standard deviation (dispersion), while categorical variables were reported as absolute frequency (“n”) and relative percentage. The Kolmogorov–Smirnov test was applied to verify the normal distribution of quantitative variables.

Categorical variables were compared using the Chi-square test, or Fisher’s exact test if any cell contained fewer than five cases. Comparisons between categorical and quantitative variables were made using Student’s *t*-test for normally distributed quantitative measures, or the Mann–Whitney U test when normality was not confirmed. To assess the diagnostic and prognostic performance of different values evaluated throughout the study, receiver-operating characteristic (ROC) curves were utilized to establish their discriminative capacity and determine the optimal cutoff value with the highest sensitivity (S) and a specificity (E) using the Youden Index.

A *p*-value of less than 0.05 was regarded as statistically significant in all analyses.

## 3. Results

### 3.1. Baseline Characteristics

A total of 329 patients were enrolled in the study. Among them, 5.8% (n = 19) developed FHCM. The global baseline characteristics of the participants are outlined in Table 2.

### 3.2. Maternal Biometric, Analytical, and Diabetological Parameters

Data related to biometric, anthropometric, and diabetological parameters are summarized in Table 3. While no significant association was identified between maternal biometric parameters and the development of FHCM, a statistically significant correlation was identified with HbA1c levels across pregestational (8.2% vs. 7.3%, *p* = 0.003), first-visit (7.6% vs. 6.9%, *p* = 0.001), last-visit (6.9% vs. 6.7%, *p* = 0.003), and post-gestational (7.5% vs. 6.6%, *p* = 0.008) determinations.

Gestation planning also exhibited a noteworthy connection with the onset of FHCM, indicating a higher incidence of planned pregnancies in cases without FHCM (27.1% vs. 5.3%, *p* = 0.035). No statistically significant differences were found regarding the insulin dosage at the beginning and end of gestation, as well as among the various subtypes of the White classification in both cohorts.

We have compiled the HbA1c measurements taken throughout pregnancy in patients carrying fetuses diagnosed with FHCM (Figure 2).

### 3.3. Fetal Biometric Outcomes

Fetal biometric data are presented in Table 4. The cohort of fetuses with FHCM exhibited a significantly higher mean birth weight, both in absolute terms (3893 g vs. 3389 g, *p* = 0.001) and when adjusted for birth weight percentile (p88 vs. p73, *p* = 0.001). Furthermore, there was a significantly higher percentage of fetuses with a mean birth weight exceeding 4000 g (63.2% vs. 17.7%, *p* < 0.001, OR 9.20, 95% CI 3.31–25.58) and exceeding 4500 g (31.6% vs. 1.6%, *p* < 0.001, OR 30.3, 95% CI 8.10–113.38). This is noteworthy despite finding a significantly lower mean gestational age at birth in the cohort of fetuses with FHCM (*p* = 0.001).

### 3.4. Performance Evaluation of Early HbA1c Level for Predicting Hypertrophic Cardiomyopathy

Considering the association between the initial visit’s HbA1c levels and the incidence of FHCM, ROC was constructed to evaluate the discriminatory ability of HbA1c in predicting the development of FHCM (Figure 3). Additionally, we aimed to identify the optimal cutoff value for HbA1c at the initial visit, effectively distinguishing pregestational diabetic pregnant women at an elevated risk of experiencing FHCM. With this determination, we obtained an AUC of 0.75 (95% CI 0.61–0.83; *p* < 0.001), suggesting that the model exhibits a reasonable capability in predicting the occurrence of FHCM.

To determine the optimal threshold that balances sensitivity and specificity in our ROC curve evaluation, we employed the Youden Index, aiming to maximize the model’s ability to discriminate between positive and negative classes. Our choice of threshold is based on maximizing the Youden Index, ensuring an optimal balance between the ability to detect true positives and the ability to avoid false positives. The HbA1c value of 7.15% sets the threshold for providing optimal discriminative power. This threshold demonstrates a sensitivity of 78.9% and specificity of 63%. Approximately 38.9% of pregnant women (n = 160), equivalent to around 23 patients per year in our environment, fall within this range.

The HbA1c value of 8.35% serves as the threshold to prioritize maximizing specificity while maintaining a reasonable level of sensitivity, strategically selected as the point on the ROC curve nearest to the (0.1) corner. The selected threshold yields a specificity of 26.3% and a sensitivity of 90.8%. Approximately 11.2% of pregnant women (n = 46), equivalent to around seven patients per year in our environment, fall within this range. The selection of this threshold strategically prioritizes high specificity, ensuring a more resource-efficient approach while maintaining reasonable sensitivity for predicting the likelihood of FHMC.

## 4. Discussion

### 4.1. Main Findings

This study aimed to determine the predictive value of HbA1c at the first gestational visit for the development of FHCM in women with PGDM. Out of 329 women included in this study, 19 cases (5.8%) resulted in fetuses prenatally diagnosed with FHCM. Pregnant women with FHCM-diagnosed fetuses exhibited higher pregestational HbA1c levels, as well as elevated HbA1c levels at the initial visit, last prenatal visit, and postnatal visit. Furthermore, fewer of these women had planned their pregnancies.

The cohort of fetuses diagnosed with FHCM displayed significantly higher mean birth weights, both in absolute terms and when adjusted for birth weight percentile. Moreover, a significantly greater proportion of fetuses had birth weights exceeding 4000 g and exceeding 4500 g, denoting a process of generalized organomegaly, culminating in higher rates of macrosomia and FHCM.

Considering the statistical relationship between HbA1c levels at different stages of pregnancy and FHCM, along with the higher incidence of macrosomia in these fetuses resulting from poorer glycemic control during gestation, we conducted ROC analysis to assess the ability to predict the occurrence of FHCM based on HbA1c at the first pregnancy visit. The analysis yielded an AUC of 0.75 (95% CI 0.61–0.83; *p* < 0.001), indicating reasonable predictive capability. We identified 7.15% as the HbA1c value with the best discriminative power. Approximately 38.9% of pregnant women (n = 160) fell within this threshold. Additionally, for resource optimization, a second threshold at 8.35% was strategically chosen as the point nearest to the (0.1) corner of the ROC curve, prioritizing high specificity (90.8%). This threshold encompassed 11.2% of pregnant women (n = 46). These findings provide valuable insights into early predictors and risk stratification for FHCM in the context of maternal PGDM.

### 4.2. Impact of Poor Glycemic Control on Fetal Hypertrophic Cardiomyopathy

Chronic hyperinsulinemia and utero–placental hypoxic events resulting from maternal PGDM can have a deleterious effect on the fetus, with the fetal heart being one of the target organs [24]. The relatively hyperglycemic intrauterine environment leads to a decrease in left ventricular preload and an increase in right ventricular afterload [22]. In this regard, our group has previously demonstrated the positive impact of gestational planning on the occurrence of congenital anomalies, including congenital heart defects and FHCM, possibly stemming from improved glycemic control in these pregnant individuals [25].

This histological impact translates into the alteration of fetal echocardiographic parameters [26]. Topcuoglu et al. compared 41 offspring of diabetic mothers with 51 controls, observing a significant increase in the thickness of the interventricular septum in diastole, the thickness of the left ventricular posterior wall in diastole, and left ventricular mass in the offspring of diabetic mothers [27].

FHCM remains a relatively frequent occurrence among the infants of diabetic mothers, even when glycemic control is well maintained [26,28,29,30,31,32,33]. Moreover, clinical guidelines recommend offering a detailed fetal echocardiography to all pregnant women with pregestational diabetes due to their increased risk of congenital heart defects [34]. However, its close relationship with hyperinsulinemia means that poor glycemic control can maximize this effect and increase the incidence of congenital heart defects. El-Ganzoury’s group found a correlation between poor maternal glycemic control (HbA1c ≥ 7.0%) and increased thickness of the interventricular septum [35], while Hernández Del Río’s group found a higher mean HbA1c at the last pregnancy check-up in mothers with offspring with FHCM (9.5% vs. 7.3%) [36]. However, no previous studies have considered the use of this parameter at the onset of pregnancy as a predictor of FHCM occurrence.

In our study, we observed a statistically significant relationship between FHCM and HbA1c levels at different pregestational (8.2% vs. 7.3%, *p* = 0.003), first-visit (7.6% vs. 6.9%, *p* = 0.001), last-visit (6.9% vs. 6.7%, *p* = 0.003), and post-gestational (7.5% vs. 6.6%, *p* = 0.008) determinations. These data allow us to link metabolic control (defined by HbA1c level) to the occurrence of FHCM. Thus, we tested its determination in the first trimester as a screening test to determine the cut-off point that defines a higher risk of having a child with FHCM linked to metabolic control at that moment of pregnancy. This enabled us to determine the cost-effectiveness of additional echocardiographic monitoring for a selected cohort of pregnant women.

### 4.3. Overall Performance of HbA1c at First Visit for Predicting FHCM

While FHCM can occur in any pregnant woman with pregestational and gestational diabetes, even with well-controlled disease [37], most cases arise in those with poor glycemic control. However, the definition of “poor glycemic control” varies widely among groups and often relies on thresholds derived from studies in non-pregnant populations rather than factors related to the study itself.

Gonzalez’s group concluded that FHCM occurs in 22% of pregnancies in mothers with PGDM, with a higher incidence in those with HbA1c > 7% compared to those with HbA1c < 5.9% in the third trimester [38]. However, the group did not determine the diagnostic accuracy of metabolic control in terms of discriminative capacity and optimal cutoff point for sensitivity and specificity.

Given this common deficiency in previous studies, we generated a ROC curve to determine the optimal cutoff value of HbA1c at the first visit to identify PGDM women at higher risk of having a fetus with FHCM. We obtained an AUC with good discriminative power (72%), with a value of 7.15% representing the best predictor of FHCM risk. This cutoff value includes approximately 38.9% of pregnant women. Only four out of nineteen patients with diagnosed FHCM had HbA1c values at the first visit below 7.15% (6.8%, 6.6%, 6.5%, and 5.4%).

To optimize resources in high-prevalence areas, the first value with specificity exceeding 90% is 8.35% (S 26.3%, E 90.8%). This corresponds to 11.2% of pregnant women. Here, based on our data, we could have potentially diagnosed five out of nineteen patients.

### 4.4. Strengths and Limitations

One of the strengths of this study lies in the substantial number of pregnant women with PGDM included. Moreover, by conducting the research at a single center, we were able to apply a consistent methodology and patient management strategy, thereby strengthening the internal validity of the results and minimizing potential confounding factors.

A limitation arises from this study’s retrospective nature; however, it should be noted that the data were collected prospectively during the entire study period. This approach reduces biases typically linked to retrospective studies and enhances the reliability of the findings. Furthermore, the relatively low incidence of FHCM constrained the statistical power to detect significant associations.

This study did not differentiate between varying FHCM degrees of severity, nor did it distinguish between cases where the condition resolved postnatally and those where it persisted over time.

## 5. Conclusions

HbA1c levels at the first gestational visit serve as predictive markers for the development of FHCM in women with PGDM. Our analysis revealed a significant correlation between HbA1c levels throughout pregnancy and the incidence of FHCM, underscoring the critical role of glycemic control during both pregestational and gestational periods.

The ROC analysis revealed a robust predictive capacity, highlighting the potential of HbA1c levels as early indicators of at-risk pregnancies. Additionally, our determination of optimal HbA1c thresholds improves risk stratification, facilitating targeted interventions and efficient resource allocation.

## Figures and Tables

**Figure 1 children-12-00312-f001:**
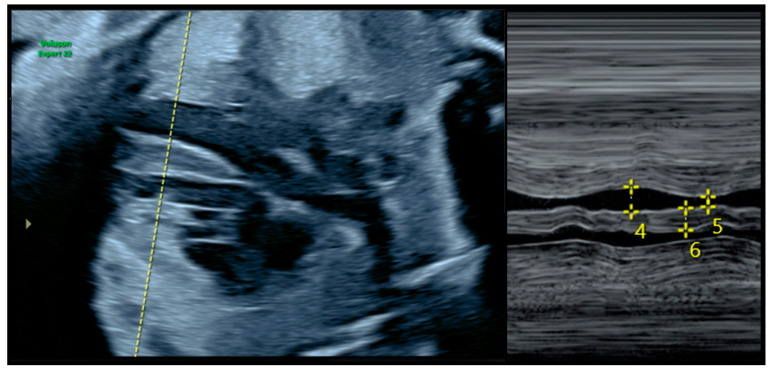
Measurement of fetal ventricular septal thickening (M-mode ultrasound). The number “6” indicated by the calipers represents the measured thickness of the ventricular septum.

**Figure 2 children-12-00312-f002:**
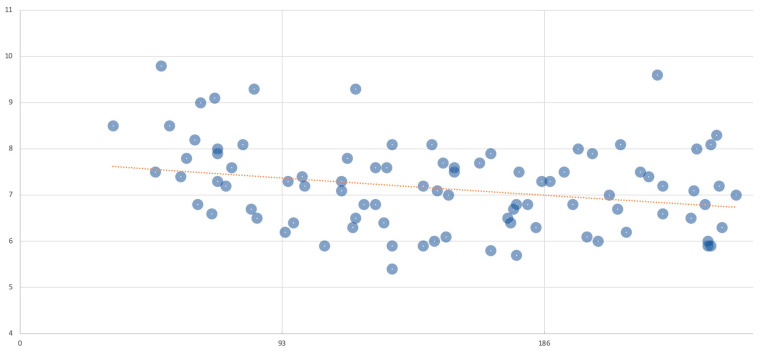
Trend of HbA1c levels (Ud) during pregnancy (days) in patients with FHCM.

**Figure 3 children-12-00312-f003:**
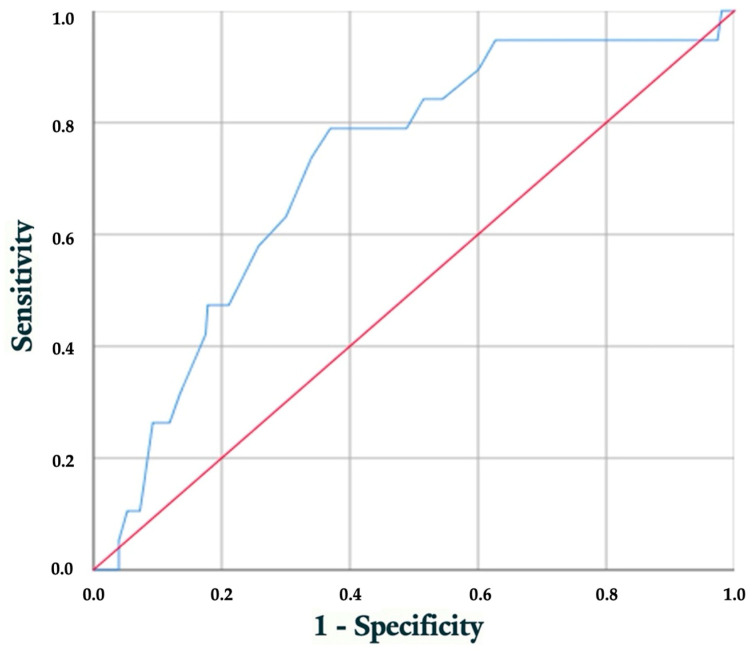
Performance evaluation of early HbA1c level for predicting FHCM. ROC curve.

**Table 1 children-12-00312-t001:** Management of pregestational diabetic pregnancies (adapted from the updated 2020 practical care guideline on diabetes mellitus and pregnancy [SEGO]).

Gestational Week/Time	Clinical Setting	Main Actions
Preconception Visit	Endocrinology Clinic	- Evaluate maternal history, vascular complications, and comorbidities - Optimize glycemic control and discontinue teratogenic medications - Ensure availability of monitoring supplies - Begin folic acid and iodine supplementation - Advise on lifestyle (smoking cessation, alcohol avoidance) - Recommend contraception until optimal metabolic control is achieved
First Visit (confirmation of pregnancy)	Diabetes/Pregnancy Clinic and Endocrinology	- Confirm pregnancy viability and accurate dating - Reassess metabolic control and adjust insulin regimen - Consider low-dose aspirin prophylaxis for preeclampsia - Continue folic acid supplementation
11.0–13.6 weeks	Obstetrics and Diabetes/Pregnancy Clinic	- Early ultrasound for fetal viability, number of fetuses, and nuchal translucency - Complete chromosomal anomaly screening - Assess maternal well-being and screen for complications - Adjust insulin therapy if needed
18.0–21.6 weeks	Fetal Medicine and Diabetes/Pregnancy Clinic	- Detailed fetal anatomical ultrasound, including fetal echocardiography - Evaluate maternal well-being and detect complications - Assess fetal growth and amniotic fluid - Continue optimizing glycemic control and monitor renal parameters (albuminuria, creatinine)
From 28.0 weeks (visit schedule depends on metabolic control: well controlled: weeks 28, 32, 36, 37, 38; poor control: weekly visits)	Diabetes/Pregnancy Clinic and Endocrinology	- Monitor maternal well-being and detect complications - Assess fetal growth, amniotic fluid volume, and placental status - Perform fetal echocardiography at each visit from ~36.0 weeks - Conduct Group B Streptococcus screening and third-trimester labs - Adjust insulin therapy as needed and maintain close metabolic control
Delivery	Hospital	- Elective induction at ≥38.6 weeks if good glycemic control and no complications - Consider earlier induction (≥36.0 weeks) if suboptimal control or maternal/fetal complications - If needed before 34.6 weeks, administer corticosteroids for fetal lung maturity and adjust insulin accordingly - Vaginal delivery is preferred; cesarean is indicated for estimated fetal weight >4500 g, previous shoulder dystocia, or standard obstetric indications

**Table 2 children-12-00312-t002:** Baseline characteristics.

Baseline Characteristics	Patients Enrolled N = 329
Maternal age, yrs	40 ± 5.22
Maternal weight at the onset of pregnancy, kg	72.4 ± 16.35
Height, cm	162 ± 6.00
Body Mass Index at the onset of pregnancy, kg/m^2^	26.0 ± 5.48
Ethnicity, n *(%)*	
Caucasian	313 (95.1%)
Black	8 (2.4%)
Maghrebi	7 (2.1%)
Asian	1 (0.3%)
Smoker, n *(%)*	64 (19.5%)
ART, n *(%)*	23 (7.0%)
Nulliparous, n *(%)*	111 (33.7%)
Previous cesarean section, n *(%)*	85 (25.8%)
Type of PGDM, n *(%)*	
Type 1 PGDM	237 (72.0%)
Type 2 PGDM	90 (27.4%)
MODY PGDM	1 (0.3%)
LADA PGDM	1 (0.3%)
Pregestational hypertension, n *(%)*	44 (13.4%)

PGDM: Pregestational Diabetes Mellitus; LADA: Latent Autoimmune Diabetes in Adults; ART: Assisted Reproductive Technology; MODY: Maturity-Onset Diabetes of the Young. Data are presented as n (%) or mean ± standard deviation.

**Table 3 children-12-00312-t003:** Maternal biometric, analytical, and diabetological parameters.

	FHCM N = 19	No FHCM N = 310	*p*-Value
Maternal weight at the onset of pregnancy, kg	70 ± 13.21	72 ± 16.26	0.526
Maternal weight at the end of pregnancy, kg	85 ± 1.05	84 ± 15.71	0.689
Body Mass Index at the onset of pregnancy, kg/m^2^	26.6 ± 5.24	27.6 ± 6.33	0.615
Body Mass Index at the end of pregnancy, kg/m^2^	32.5 ± 5.48	32.4 ± 6.05	0.721
Pregestational HbA1c (%)	8.2 ± 1.00	7.3 ± 1.32	0.003
HbA1c at first visit (%)	7.6 ± 0.92	6.9 ± 1.12	0.001
HbA1c at last visit (%)	6.9 ± 1.03	6.7 ± 0.78	0.003
HbA1c increment (%)	−0.74 ± 0.78	−0.73 ± 1.03	0.677
Postgestational HbA1c (%)	7.5 ± 0.60	6.6 ± 1.13	0.008
Planned pregnancy	1 (5.26%)	84 (27.1%)	0.035
White’s Classification			0.602
Insulin dosage at first visit (Ud)	42 ± 15.08	42 ± 21.53	0.783
Insulin dosage at last visit (Ud)	69 ± 28.91	67 ± 32.17	0.425
Insulin increment (Ud)	29 ± 24.78	28 ± 23.97	0.961

Data are presented as n (%) or mean ± standard deviation.

**Table 4 children-12-00312-t004:** Fetal biometric outcomes.

	FHCM N = 19	No FHCM N = 310	*p*-Value	OR (95% CI)
Birth weight, grams	3893 ± 1006	3389 ± 651	0.001	-
Weight > 4000 g	12 (63.2%)	55 (17.7%)	<0.001	9.20 (3.31–25.58)
Weight > 4500 g	6 (31.6%)	5 (1.6%)	<0.001	30.3 (8.10–113.38)
Weight percentile	88 ± 29.51	73 ± 31.15	0.001	-
Gestational age at birth, weeks	36.5 ± 2.11	37.6 ± 1.66	0.001	-
Birth < 37 weeks	7/18 (38.9%)	61/307 (19.9%)	0.054	2.57 (0.96–6.89)

Data are presented as n (%) or mean ± standard deviation.

## Data Availability

The datasets used during this current study are available from the corresponding author on reasonable request. The data are not publicly available due to privacy and legal reasons.

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
