# Peer review of "Predictive Value of Maternal HbA1c Levels for Fetal Hypertrophic Cardiomyopathy in Pregestational Diabetic Pregnancies"

_children, 2025, doi:10.3390/children12030312_

Round 1
Reviewer 1 Report
Comments and Suggestions for Authors
Dear authors
congratulations
it has been a hige pleasure for me to read and review your manuscript
The topic is of interest the study well conducted and the manuscript well presented
i have minor revisions to suggest
1) please improve the introduction mentioning about the risk of stillbirth associated to pregestational diabetes whom increase at increasing Hb1Ac levels, there are evidence to support the relationship between not controlled diabetes and LGA with stillbirth (read and cite doi: 10.36129/jog.2022.20)
2) moreover mention to the guidelines that recommedn a fetal echocardiography to all women affected by pregestational diabetes due to their higher risk of CHDs
Best regards
Author Response
1. Please improve the introduction mentioning about the risk of stillbirth associated to pregestational diabetes whom increase at increasing Hb1Ac levels, there are evidence to support the relationship between not controlled diabetes and LGA with stillbirth (read and cite doi: 10.36129/jog.2022.20)
1. In accordance with your suggestion, we have revised the introduction to address the increased risk of stillbirth associated with rising HbA1c levels in pregnancies complicated by pregestational diabetes. We have also included a citation to support the relationship between inadequate glycemic control, large-for-gestational-age (LGA) fetuses, and stillbirth (doi: 10.36129/jog.2022.20).
2. Moreover mention to the guidelines that recommedn a fetal echocardiography to all women affected by pregestational diabetes due to their higher risk of CHDs
2. In response to your suggestion, we have now included a reference to the guidelines advising that all women with pregestational diabetes undergo a detailed fetal echocardiography, as they are at higher risk for congenital heart defects. We have cited the relevant guidelines (NICE NG3), which address the management of diabetes from preconception through the postpartum period.
Reviewer 2 Report
Comments and Suggestions for Authors
I have the following comments:
- In section 2.2 entitled diagnosis of FHCM and echocardiographic parameters the authors state the following: “The diagnosis of FHCM was primarily based on interventricular septal thickening exceeding 2 standard deviations relative to the mean for gestational age (reference 18 Figure 1)”. Reference 18 is a meta-analysis of a number of studies which measured the interventricular septal thickness in the second and third trimesters of pregnancy. While the individual studies reported the mean and standard deviation between the control group and those with pregestational or gestational diabetes, there was no reported average of the mean and standard deviation. Therefore, the authors of the current submission should provide the mean and standard deviation they used to determine whether the interventricular septal thickness was increased or not. I would suggest they provide a graph illustrating the mean and standard deviation of the interventricular septal thickness that they used and plot the individual values for the patients in this study. This would assist the reader to visualize the distribution of septal thickness which was used to base their analysis of the hemoglobin A1C values.
- The authors measured the interventricular septal fitness at what appears to be the mid -portion of the septum. However, because the septum is shaped like a triangle with the apex at the insertion of the AV valves and the base at the apex of the heart, the septal thickness can vary based upon where the M mode is placed. A newer technique which measures the area of the interventricular septum demonstrated that irrespective of whether the measurement was made at end- diastole or end-systole, the areas were the same. Since the authors had a lower incidence than reported in other studies of interventricular septal hypertrophy, they may want to consider reanalyzing their data set by measuring the area of the interventricular septum and computing the Z score using the calculator provided as a supplement in the following reference. The calculator also allows the examiner to not only measure the area but also the mid septal thickness for comparison.
Measuring the Area of the Interventricular Septum in the 4-Chamber View: A New Technique to Evaluate the Fetus at Risk for Septal Hypertrophy
Greggory R DeVore 1 2 3, Berthold Klas 4, Gary Satou 5, Mark Sklansky 5
J Ultrasound Med . 2022 Dec;41(12):2939-2953. DOI: 10.1002/jum.15980
Abstract
Objectives: One of the problems for the clinician who desires to measure the interventricular septum (IVS) in a high-risk fetus is to know where to make the measurement. The purpose of this study was to use speckle-tracking analysis to measure the IVS area, 24-segment widths, and length at end-diastole (ED) and end-systole (ES) in normal fetuses.
Methods: From the 4-chamber view, speckle-tracking analysis was performed at ED and ES on the IVS in 200 normal fetuses. The following were computed and regressed against gestational age (GA) and fetal biometric (FB) measurements: area, length, and the 24-segment transverse widths from the apex to the crux. The 24-segment width/length ratio was also measured. The speckle-tracking measurements of the ED area and length were compared using a point-to-point measurement tool available on all ultrasound machines.
Results: The ED and ES areas, lengths, and 24-segment widths increased with GA and FB. The ED and ES areas were virtually identical. The 24-segment width/length ratio decreased from the apex to the crux of the septum. There was no significant difference in the measurement of the ED area and the length between speckle-tracking and the point-to-point measurements.
Conclusions: Measurement of the area and length of the IVS are simple to obtain and provide a new diagnostic tool to evaluate the fetus at risk for IVS hypertrophy which may be observed in fetuses of mothers with pregestational and gestational diabetes.
Author Response
1. In section 2.2 entitled diagnosis of FHCM and echocardiographic parameters the authors state the following: “The diagnosis of FHCM was primarily based on interventricular septal thickening exceeding 2 standard deviations relative to the mean for gestational age (reference 18 Figure 1)”. Reference 18 is a meta-analysis of a number of studies which measured the interventricular septal thickness in the second and third trimesters of pregnancy. While the individual studies reported the mean and standard deviation between the control group and those with pregestational or gestational diabetes, there was no reported average of the mean and standard deviation. Therefore, the authors of the current submission should provide the mean and standard deviation they used to determine whether the interventricular septal thickness was increased or not. I would suggest they provide a graph illustrating the mean and standard deviation of the interventricular septal thickness that they used and plot the individual values for the patients in this study. This would assist the reader to visualize the distribution of septal thickness which was used to base their analysis of the hemoglobin A1C values.
1. In response to your suggestion, we have revised the manuscript to clarify that the diagnosis of fetal hypertrophic cardiomyopathy (FHCM) was based on septal thickness exceeding two standard deviations above the mean for gestational age. We have defined the normal reference values for septal thickness using the standards established by Schneider et al. via the Fetal Medicine Barcelona online calculator (https://fetalmedicinebarcelona.org/calc/).
We acknowledge that a more detailed illustration of IVS distribution could be valuable in clarifying the cutoff we employed. However, the primary objective of our study was not to establish or re-validate normative values for fetal interventricular septal measurements but rather to evaluate how maternal glycemic control impacts the development of FHCM. Consequently, we focused on the relationship between different levels of hemoglobin A1C and the resultant myocardial changes, rather than on redefining the precise echocardiographic reference ranges or pooling mean/SD data from multiple studies.
Although we considered presenting an additional figure specifically on the distribution of IVS measurements, we ultimately concluded that it might overemphasize an aspect that is supplementary to the main objective of examining the glycemic control thresholds for FHCM risk. Nonetheless, we fully recognize the importance of clear echocardiographic reference standards, and we agree that future investigations exclusively dedicated to the standardization of fetal IVS measurements could address this point more comprehensively.
We appreciate your constructive feedback and trust that these revisions address your concerns.
2. The authors measured the interventricular septal fitness at what appears to be the mid -portion of the septum. However, because the septum is shaped like a triangle with the apex at the insertion of the AV valves and the base at the apex of the heart, the septal thickness can vary based upon where the M mode is placed. A newer technique which measures the area of the interventricular septum demonstrated that irrespective of whether the measurement was made at end- diastole or end-systole, the areas were the same. Since the authors had a lower incidence than reported in other studies of interventricular septal hypertrophy, they may want to consider reanalyzing their data set by measuring the area of the interventricular septum and computing the Z score using the calculator provided as a supplement in the following reference. The calculator also allows the examiner to not only measure the area but also the mid septal thickness for comparison.
2. Thank you for this highly relevant and valuable suggestion. We agree that measuring the interventricular septal area is of great interest and intend to incorporate this technique in future studies and in our routine practice. However, after reviewing each case included in our study—both those with FHCM and those without—we found that we lack sufficient image captures and the necessary quality to retrospectively assess the septal area. Furthermore, most published studies on this topic measure septal thickness rather than septal area, as this latter parameter is a relatively recent consideration. Nonetheless, we recognize the importance of your observation and will certainly keep it in mind as we continue our investigations. Thank you once again for your insightful comment.
Reviewer 3 Report
Comments and Suggestions for Authors
Dear Authors, I read with pleasure your paper that seems conceptually correct to me: finding parameters that allow us to identify early patients affected by PGDM with greater risks, as in this case of FHCM. Gestational diabetes will be, as you highlighted, an increasingly frequent problem in the management of our pregnant women. I believe that in order to better interpret the role of HcA1, it could be useful:
- a table of weight gain, of the dosage of HcA1 in the II and III trimester for the 19 patients.
- a description also with a table of what your internal protocol provides for in the management of diabetic patients.
- a description, always with a help table, that describes the degree of adherence and respect of how many and which patients have adhered to your protocol?
- were the therapy and lifestyles adequate in the patients of the entire court?
I believe that the value of pregestational HcA1 and in the first visit in the first trimester is predictive of the difficulty we will encounter in the management of some patients with PGDM not only for FHCM but for all the complications associated with this pathology.
Author Response
1. A table of weight gain, of the dosage of HcA1 in the II and III trimester for the 19 patients.
1. We have prepared a detailed table showing the weight gain for each of the 19 participants. However, after careful consideration, we feel that our manuscript already contains a substantial number of tables and figures, and that the absolute or relative weight gain data would not contribute additional meaningful insight to our findings.
Regarding the HbA1c levels during pregnancy, we have created a dedicated figure illustrating the trend of each measurement for the 19 patients, which we have now included in the manuscript. We trust this figure will provide a clear overview of the glycemic control patterns throughout pregnancy.
2. A description also with a table of what your internal protocol provides for in the management of diabetic patients.
2. Thank you for your suggestion We have now added the requested table (Table 1) to the manuscript, summarizing the key elements of our standard care and follow-up procedures.
3. A description, always with a help table, that describes the degree of adherence and respect of how many and which patients have adhered to your protocol?
3. In this study, we did not evaluate adherence as a specific outcome. Our internal protocol is intended as a guideline for healthcare professionals involved in the management of pregestational diabetic pregnancies, aiming to standardize clinical practice. As mentioned previously, we have included the protocol in the manuscript for clarification and completeness. However, it is important to note that the protocol serves as guidance for clinicians rather than a strict standard for patient adherence.
In clinical practice, we provide dietary and treatment recommendations, and, as is typical in hospital settings, individual patients may follow these guidelines with varying levels of discipline. Nonetheless, we did not assess this parameter on a patient-by-patient basis, as it was not within the scope of the current study. We do not believe the absence of an adherence measure undermines our hypothesis or the conclusions presented.
4. Were the therapy and lifestyles adequate in the patients of the entire court?
I believe that the value of pregestational HcA1 and in the first visit in the first trimester is predictive of the difficulty we will encounter in the management of some patients with PGDM not only for FHCM but for all the complications associated with this pathology.
4. As mentioned in our previous response, all patients received individualized treatment, dietary measures, and lifestyle recommendations, which were re-evaluated and adjusted at each prenatal visit. Therefore, we consider these recommendations to have been both appropriate and thoroughly assessed throughout each pregnancy.
We agree with your observation that pregestational and first-trimester HbA1c levels can serve as a predictor of overall pregnancy complications in women with pregestational diabetes mellitus (PGDM), as they reflect glycemic control during the critical early stage of embryonic development. Although we have additional data supporting this concept, as yet unpublished, it falls outside the scope of the current study. Our focus here is specifically on identifying the factors associated with the development of fetal hypertrophic cardiomyopathy and establishing a predictive threshold to determine higher-risk profiles.
Thank you for your constructive feedback.
Round 2
Reviewer 3 Report
Comments and Suggestions for Authors
The paper seems to better clarify the role and usefulness of HbA1c dosage especially in the first trimester of patients with PGDM.
Congratulations.